

# Controlled chamber formation of per- and polyfluoroalkyl substances (PFAS) aerosols with *Pseudomonas fluorescens*: size distributions, effects, and inhalation deposition potential

Ivan Kourtchev[1], Steve Coupe[1], Allison Buckley[2], Jishnu Pandamkulangara Kizhakkethil[1], Elena Gatta[3], Dario Massabò[3,4], Paolo Prati[3,4], Virginia Vernocchi[4], and Federico Mazzei[3,4]

[1]Centre for Agroecology Water and Resilience (CAWR), Coventry University, Wolston Lane, Ryton on Dunsmore, CV8 3LG, UK
[2]Toxicology Department, Radiation, Chemical, Climate and Environmental Hazards Directorate, UK Health Security Agency, Harwell Campus, Chilton, Oxfordshire, OX11 0RQ, UK
[3]Dipartimento di Fisica, Università di Genova, Genoa, Italy
[4]Istituto Nazionale di Fisica Nucleare (INFN), Sezione di Genova, Genoa, Italy

*Correspondence to*: Ivan Kourtchev (ivan.kourtchev@coventry.ac.uk)

**Abstract.** Per- and polyfluoroalkyl substances (PFAS) are recognised as atmospheric contaminants, yet processes governing their aerosol formation, size distribution, and interactions with atmospheric particle surfaces remain unknown. We investigated aerosolisation and size-resolved behaviour of 25 PFAS covering short-, medium-, and long-chain perfluoroalkyl carboxylic acids (PFCA), perfluoroalkane sulfonates, fluorotelomer sulfonates and emerging alternatives. Experiments were conducted under controlled chamber conditions using a water–organic solvent system, in the absence/presence of the model bacterium *Pseudomonas fluorescens* seed, representative of wastewater-impacted environments. Most PFAS exhibited unimodal mass–size distributions peaking at 0.3 µm, indicating dominant association with the fine mode. Sulfonated PFAS showed broadly similar aerosol-phase concentrations regardless of carbon-chain length, whereas PFCA displayed increasing aerosolisation with chain length. Perfluorooctane sulfonic acid (PFOS) showed additional ultrafine enrichment, 6:2 fluorotelomer sulfonate (6:2 FTS) and sodium 4,8-dioxa-3H-perfluorononanoate (NaDONA) exhibited broader size profiles, suggesting compound-specific effects linked to volatility and interfacial behaviour. *Pseudomonas fluorescens* seed did not enhance PFAS aerosol concentrations through condensation or heterogeneous uptake onto bacterial particles or shift in modal diameters, and no enrichment was observed at bacterial size mode, indicating limited PFAS–bioaerosol association under the tested conditions. Multiple-Path Particle Dosimetry (MPPD) modelling based on the measured size distributions predicted substantial deposition of the aerosol-bound PFAS in the pulmonary region, particularly for compounds enriched in ultrafine particles. Our findings indicate that PFAS aerosol behaviour in mixed-solvent systems is controlled primarily by physical droplet generation and evaporation, with implications for airborne transport and inhalation exposure from contaminated aqueous sources.



## 1 Introduction

Per- and polyfluoroalkyl substances (PFAS) are synthetic organofluorine compounds widely used in industrial and consumer applications due to their high thermal stability and strong surface-active behaviour (Glüge et al., 2020). Their persistence, mobility, and potential toxicity have raised growing environmental and health concerns, particularly given their ubiquitous presence in water, soil, and the atmosphere (see reviews Evich et al., 2022; Faust, 2023). Atmospheric transport is now recognised as an important pathway for the redistribution of PFAS on regional and global scales (Barber et al., 2007; Ellis et al., 2004; Kourtchev et al., 2024; Kourtchev et al., 2025; Schenker et al., 2008), yet the mechanisms by which these compounds enter and are stabilised in the atmospheric particle phase remain poorly understood.

PFAS are surfactants that readily accumulate at air–water interfaces (Schaefer et al., 2019) and adsorb onto mineral (Alves et al., 2020; Folorunsho et al., 2025), organic (Wanzek et al., 2023), and biological surfaces (Dai et al., 2023). Their strong interfacial activity suggests that pre-existing airborne particles could act as carriers or "seeds" for PFAS, facilitating their transfer from aqueous systems into the atmosphere. Laboratory studies with sea-spray aerosol have shown that PFAS enrichment in surface films exceeds that of bulk water by several orders of magnitude (Johansson et al., 2019), supporting the likelihood of interfacial transfer. In the atmosphere, aerosol particles can grow through the condensation of semi-volatile compounds when ambient vapour pressures fall below equilibrium values, leading to vapour uptake by existing particles (Romakkaniemi et al., 2011; Stolzenburg et al., 2018). The rate and extent of condensation depend on compound volatility, surface tension, and relative humidity, as well as particle composition. For surface-active species such as PFAS, adsorption or condensation at particle interfaces may not only contribute to particle growth but also promote their association with pre-existing aerosols, effectively turning these particles into carriers for atmospheric transport. Interactions with organic or biological material may further modify these processes by altering surface energy, hygroscopicity, and the stability of particle-bound PFAS. Wastewater treatment plants (WWTPs) are of particular interest in this context, as they are recognised co-sources of both PFAS (Cookson & Detwiler, 2022) and biological aerosols, also known as bioaerosol (Li et al., 2016; Poopedi et al., 2025; Xu et al., 2020). During aeration and mechanical agitation, fine droplets containing PFAS and microbial material can become airborne, providing a direct mechanism for PFAS transfer from water to air. Elevated concentrations of PFAS are frequently detected in WWTP effluents and sludges (Cookson & Detwiler, 2022), and both PFAS and bioaerosol emissions from aeration tanks are well documented (Ahrens et al., 2011; Li et al., 2016; Lin et al., 2022; Pandamkulangara Kizhakkethil et al., 2025; Poopedi et al., 2025; Shoeib et al., 2016; Xu et al., 2020).

Despite growing evidence for PFAS volatilisation and enrichment during water-to-air transfer (Ahrens et al., 2011; Lin et al., 2022; Pandamkulangara Kizhakkethil & Kourtchev, 2025; Pandamkulangara Kizhakkethil et al., 2024; Shoeib et al., 2016), the potential involvement of biological aerosol particles in facilitating PFAS aerosolisation remains unexplored. Literature indicates that bioaerosols can participate in heterogeneous chemical processes (e.g., Ervens & Amato, 2020; Estillore et al., 2016), suggesting that similar behaviour may occur for PFAS. Recent work has shown that PFAS can associate with bacterial cells, primarily through adsorption to cell surfaces and, to a lesser extent, through limited uptake into cell interiors (Dai et al.,



2023). Controlled batch and miscible-displacement studies using *Pseudomonas aeruginosa* (Gram-negative) and *Bacillus*
*subtilis* (Gram-positive) demonstrated substantial retention and increased retardation of perfluorooctane sulfonic acid (PFOS),
one of the most studied PFAS, in bacterial-inoculated porous media. While these observations are derived from aqueous and
porous-media systems and cannot be directly extrapolated to the atmosphere, they provide a useful conceptual analogue: if
bacterial cell surfaces in water promote PFAS association, analogous surface-mediated interactions could, in principle, occur
with airborne biological particles. To date, however, no studies have examined PFAS interactions with bioaerosols in the
atmospheric context, highlighting a clear and unaddressed research gap. In principle, microbial cells or their fragments could
either enhance PFAS partitioning into the particle phase through sorptive or condensation processes or inhibit it via electrostatic
or interfacial competition.
Understanding whether and how biological matter influences PFAS aerosol behaviour is therefore essential for accurately
assessing emission pathways from engineered systems such as WWTPs. Furthermore, PFAS–bioaerosol interactions may alter
the hygroscopicity and atmospheric lifetime of emitted particles, with implications for transport and deposition.
A further uncertainty concerns the atmospheric transport, deposition, and human exposure potential of PFAS, all of which are
intrinsically linked to particle size. Particle size governs the residence time and dispersal range of aerosols (Finlayson-Pitts
and Pitts, 2000), their efficiency of dry and wet deposition (Farmer et al., 2021) and likelihood of respiratory uptake (Tsuda et
al., 2013). Smaller submicron particles (<1 µm) have low gravitational settling velocities and long atmospheric residence
times, allowing efficient transport over long distances, whereas coarse particles are more readily removed from the atmosphere
near emission sources through sedimentation and impaction (Zhang and Vet, 2006). Particle size also governs deposition
behaviour and associated health outcomes: fine and ultrafine particles can penetrate deep into the pulmonary region of
respiratory systems and reach the alveoli, where they are linked to cardiovascular and respiratory morbidity (Pope III &
Dockery, 2006), whereas coarse particles deposit mainly in the upper airways and have been shown to induce inflammation
and allergic responses (Wu et al., 2018).
Despite its importance, size-resolved information on PFAS in aerosols remains scarce and not always in agreement (Dreyer et
al., 2015; Ge et al., 2017; Guo et al., 2018; Harada et al., 2006; Lin et al., 2022). Some of these studies considered a limited
number of PFAS, typically a few legacy compounds such as perfluorooctanoic acid (PFOA) and PFOS or employed samplers
with only a few size fractions (e.g. n=5), which makes direct comparison with other studies difficult. For example, Dreyer et
al. (2015) studied aerosol size distributions in a semi-rural area of Geesthacht, Germany, using a Berner-type cascade impactor
(0.14–11.4 µm) and found PFOA mainly associated with ultrafine/submicron particles (< 0.14 µm; within the fine fraction),
while PFOS showed enrichment in the supermicron range, particularly 1.4–3.8 µm. Ge et al. (2017) examined indoor and
roadside aerosols in Tsukuba, Japan, using a five-stage nano-sampler, and reported ionic perfluoroalkyl carboxylic acids
(PFCA) mainly associated with fine particles (<0.5 µm) indoors, while PFOS was enriched in coarse roadside particles (2.5–
10 µm). Guo et al. (2018) investigated urban aerosols in Shanghai, China, during a haze period using an eight-stage air sampler,
and observed bimodal distributions, with PFOA peaking in both fine (0.4–2.1 µm) and coarse (3.3–10 µm) modes, and PFOS
largely associated with coarse particles. Lin et al. (2022) analysed aerosols near wastewater treatment plants and a landfill in



Hong Kong, China, using an eleven-stage MOUDI impactor (0.056–10 µm), and found site-dependent patterns:
perfluorobutanoic acid (PFBA) and PFOA showed variable fine- and coarse-mode enrichment, perfluorobutanesulfonic acid
(PFBS) was generally coarse-dominated, and PFOS consistently peaked in the 1–10 µm range.
These studies provide important data on PFAS size distribution in aerosols; however, the available information remains too
limited to establish general patterns or identify the controlling mechanisms. Controlled laboratory investigations, though
constrained by simplified conditions and the absence of real-world variability, are therefore needed to disentangle the effects
of PFAS molecular structure and interfacial behaviour from the physical processes governing aerosol formation and droplet
drying. Such information is critical for improving atmospheric fate models and exposure assessments.
The aim of this study was to advance understanding of PFAS aerosol formation and size distribution under controlled
laboratory conditions, with particular focus on the potential role of bioaerosols as carriers. In this study, we explore how PFAS
with varying carbon chain lengths and functional groups undergo aerosolisation, both in the absence and presence of the model
bacterium *Pseudomonas fluorescens*. This bacterium was chosen as it is commonly detected in wastewater and wastewater-
impacted matrices, having been isolated from WWTP influent and effluent (including species-level recovery of *Pseudomonas*
*fluorescens*) and from raw sewage of municipal treatment plants (phage isolation targeting *Pseudomonas fluorescens*)
(Luczkiewicz et al., 2015; Sillankorva et al., 2008). Moreover, it presents a low biosafety risk, being classified as a non-
pathogenic, Risk Group 1 organism suitable for use in controlled laboratory experiments. Size-resolved PFAS aerosol
concentrations were determined for both systems to assess the influence of molecular structure and biological material on
particle-phase behaviour. To the best of our knowledge, this is the first study to directly investigate PFAS–bioaerosol
interactions and resolve their mass size distribution from polar solvent systems under controlled laboratory conditions. The
resulting mass–size distributions were then applied to the Multiple-Path Particle Dosimetry (MPPD) model to evaluate how
aerosol size affects potential respiratory deposition and human exposure. Such modelling provides a quantitative context for
interpreting the potential health relevance of observed PFAS size distributions.
**2 Materials and methods**
**2.1 Experimental facility, setup, and conditions**
The experiments were performed in the Chamber for Aerosol Modelling and Bio-aerosol Research (ChAMBRe) facility at the
University of Genoa, Italy. The chamber is a stainless-steel vessel with an internal volume of 2.2 m³, designed for studies of
particle generation, ageing, and interaction under controlled environmental conditions (Massabò et al., 2018). All the
experiments were conducted at ambient pressure and in dark conditions. Temperature and humidity inside ChAMBRe were
continuously monitored and maintained at 23 ± 3 °C and 40 ± 6 %, respectively.
Before each experiment, ChAMBRe was evacuated using a composite pumping system (rotary and root pumps) to achieve an
internal pressure of approximately $5 \times 10^{-2}$ mbar.  The reestablishment of atmospheric pressure was facilitated by introducing
ambient air into the chamber using a five-stage filtration, purification, and drying intake system, which comprised an absolute



HEPA filter and a zeolite trap (Vernocchi et al., 2023). A Waveband Integrated Bioaerosol Sensor (WIBS-NEO, Droplet
Measurement Technologies®) has been incorporated into the ChAMBRe particle monitoring system to quantify bio-aerosol
concentration. The extensive data produced by the WIBS during the ChAMBRe experiments were analysed using custom
software developed in Igor 8.0 (Wavemetrics, Inc.), designed to extract airborne bacteria/bioaerosol concentration and size
distribution within the chamber as a function of time and fluorescence intensity. In parallel, total particle number and size
distributions were monitored in real time using a Scanning Mobility Particle Sizer (SMPS 3938, TSI Inc.) equipped with a
differential mobility analyser (DMA 3081A) and a condensation particle counter (CPC 3750) in the range from 18 to 500 nm
and an Optical Particle Sizer (OPS, TSI 3330) covering 0.3–10 µm range.

**2.2 Aerosol generation and introduction**

**2.2.1 PFAS-only experiments**

A mixed standard solution containing 25 PFAS was prepared using the EPA-533PAR native analyte mixture supplied by
Wellington Laboratories (Ontario, Canada). The mixture comprised a broad suite of ionic PFAS, including perfluoroalkyl
carboxylic acids (PFCA; C4–C12), perfluoroalkane sulfonates (PFSA; C4, C5, C7 linear, and both linear and branched isomers
of C6 and C8), and several fluorotelomer sulfonates and emerging replacement compounds. Specifically, the analyte mixture
consisted of: 4:2 fluorotelomer sulfonate (4:2 FTS); 6:2 fluorotelomer sulfonate (6:2 FTS); 8:2 fluorotelomer sulfonate (8:2
FTS); hexafluoropropylene oxide dimer acid (HFPO-DA); perfluoro(2-((6-chlorohexyl)oxy)ethanesulfonic acid) (9Cl-
PF3ONS); perfluoro(2-ethoxyethane)sulfonic acid (PFEESA); perfluoro-3-methoxypropanoic acid (PFMPA); perfluoro-3,6-
dioxaheptanoic acid (3,6-OPFHpA); perfluoro-4-methoxybutanoic acid (PFMBA); perfluorobutane sulfonic acid (L-PFBS);
perfluorobutanoic acid (PFBA); perfluorodecanoic acid (PFDA); perfluorododecanoic acid (PFDoA); perfluoroheptane
sulfonic acid (L-PFHpS); perfluoroheptanoic acid (PFHpA); perfluorohexane sulfonic acid (PFHxS); perfluorohexanoic acid
(PFHxA); perfluorooctane sulfonic acid (PFOS); perfluorooctanoic acid (PFOA); perfluorononanoic acid (PFNA);
perfluoropentane sulfonic acid (L-PFPeS); perfluoropentanoic acid (PFPeA); perfluoroundecanoic acid (PFUdA); sodium
dodecafluoro-3H-4,8-dioxanonanoate (NaDONA); 11-chloroeicosafluoro-3-oxaundecane-1-sulfonic acid (11Cl-PF3OUdS).
All compounds were diluted to a final concentration of 0.5 ng mL⁻¹ in a mixture of 40:60 (v/v) methanol and ultrapure water
(18.2 MΩ cm) to ensure adequate solubility and minimise losses to container surfaces.
The addition of methanol (≥99.9% (GC), LiChrosolv®, Supelco) was necessary to prevent analyte loss due to sorption onto
the nebuliser container walls and to improve the solubility of long-chain PFAS in water. Aerosols were generated using a
three-jet Collison nebuliser operated at 5 L/min and introduced into ChAMBRe through a stainless-steel inlet connected
directly to the chamber. Aerosol generation continued for 30 min, followed by a mixing period of 10 min before sampling,
facilitated by the mixing fan installed at the base of the ChAMBRe. The internal fan was operated at 5 Hz, a setting shown by
Massabò et al. (2018) to achieve complete mixing in the ChAMBRe within approximately 2 min. Experiments were repeated
three times and are referred to as Exp 1-3 (no bacteria) below. Aerosol drying prior to chamber introduction was intentionally





avoided to minimise PFAS losses. PFAS are known to interact with surfaces and can partition during drying, so passing the
aerosol through additional tubing or drying devices (e.g., diffusion dryers/denuders) would introduce unnecessary interfaces
and increase the risk of losses. Introducing the wet aerosol directly into the chamber therefore ensured that the measured
composition reflected primary aerosol generation rather than processing artefacts.

### 2.2.2 PFAS and *Pseudomonas fluorescens* experiments

The Pseudomonas fluorescens ATCC 13525 (obtained from the American Type Culture Collection, University Boulevard,
Manassas, Virginia, United States) was grown in 30 mL volume of nutrient broth medium. The culture was incubated at 25°C
with continuous shaking in a shaker incubator (SKI 4 ARGOLAB, Carpi, Modena, Italy) at 200 rpm. The growth curve was
monitored by measuring the absorbance at $\lambda$ = 600 nm using a spectrophotometer (Shimadzu 1900) until it reached the
stationary phase (approximately 1) corresponding to about 109 cells mL$^{-1}$. Subsequently, 20 mL of the bacterial suspension
was centrifuged at 5000 rpm for 10 min, and the cell pellet was resuspended in 20 mL of sterile Milli Q (MQ) water.
The bacteria in MQ were nebulised using Sparging Liquid Aerosol Generator (SLAG) (CH Technologies, USA) with a 0.75"
diameter porous disc and nominal pore size of 2 $\mu$m. 3 mL of the bacterial suspension with a syringe pump flowrate of 0.4 mL
min$^{-1}$ were dripped onto the SLAG porous stainless-steel disk and nebulised inside ChAMBRe with a flowrate of 3.5 lpm as
performed in previous experiments (Gatta et al., 2025).
Bioaerosol was allowed to mix for 5 min, followed by addition of PFAS in the same way as described in 2.2.1. Aerosol
generation continued for 30 min, followed by a mixing period of 10 min before sampling. Experiments were repeated three
times and are referred to as Exp 4-6 (with bacteria) below.

### 2.2.3 Blank experiments

Blank experiments were conducted to assess potential contamination or background levels arising from the experimental setup
and to correct for any systematic bias in the measurements. A 40:60 (v/v) methanol–ultrapure water mixture without PFAS
was nebulised using a Collison nebuliser at 5 L min$^{-1}$. Aerosol generation was maintained for 30 min, followed by a 10 min
mixing period prior to sampling. All other experimental conditions were identical to those described in Section 2.2.1. The
blank experiments were repeated three times.

### 2.3 Aerosol generation and introduction

Size-segregated aerosol samples were collected using a Nano Micro-Orifice Uniform Deposit Impactor (NanoMOUDI-IITM,
Model 125B, MSP Corporation, USA) operated at a flow rate of 30 L min$^{-1}$. The NanoMOUDI provided aerodynamic cut-off
diameters of 10.000, 5.600, 3.200, 1.800, 1.000, 0.560, 0.320, 0.180, 0.100, 0.056, 0.032, 0.018, and 0.010 $\mu$m.
Total suspended particles (TSP) were collected in parallel with a double cone sampler, directly connected to ChAMBRe, at a
flow rate 10 L min$^{-1}$. The total sampling duration for both NanoMOUDI and TSP was 2 hours.



PallFlex 2500 QAO-UP quartz fibre filters were used as substrates in both samplers. Due to the unavailability of PallFlex 2500
QAO-UP filters for all experiments, quartz microfibre filters (RVMSFQ47Q90, Mega Systems s.r.l.) were used in TSP
sampling in one of the replicate experiments involving bacteria. All filters were prebaked at 450 °C for 2.5 hours prior to
sampling. Based on recoveries and variability between TSP replicates (RSD < 10%), which was not higher than that observed
for replicates using the same filter type, the use of a different filter brand is expected to have a negligible impact on the results.
Following sampling, aerosol samples were wrapped into aluminium foil (prebaked at 450 °C for 2 hours) avoiding contact
with any plasticware and external environment and stored at –20 °C until extraction.

## 2.4 PFAS extraction and analysis

Samples generated in the chamber were extracted and analysed following the procedure described by Kourtchev et al. (2022).
Briefly, filter edges in contact with the sampler gaskets were removed prior to extraction. Each filter was placed in a precleaned
10 mL glass vial (Chromacol 10-HSV, Thermo Scientific) and spiked with 25 µL of an internal standard (IS) mixture
containing 16 isotopically labelled ($^{13}$C) PFAS at 1 ng mL$^{-1}$ and three telomer sulfonates (M2-4:2 FTS, M2-6:2 FTS, and M2-
8:2 FTS) at 4 ng mL$^{-1}$. Full names of the isotopically labelled PFAS are shown in Table S1 of supplement. Samples were
extracted twice with 5 mL of Liquid Chromatography Mass Spectrometry (LC-MS, CHROMASOLV TM ≥99.9%) grade
methanol using ultrasonic agitation in a chilled water bath for a total of 40 min (2 × 20 min). The combined extracts were
filtered through PTFE membrane filters (0.45 µm, Iso-Disc PTFE-13-4) into prewashed 10 mL glass vials (Chromacol 10-
HSV) tightly closed with metal screw caps and PTFE septa (Chromacol 18-MSC and 18-ST101).
Extracts were then reduced in volume to 1 mL under a gentle stream of nitrogen and stored at 4 °C until analysis. On the day
of analysis, 4 mL of Optima™ LC-MS grade water was added to each sample, which was subsequently vortex-mixed and
analysed using online SPE consisting of an EQuan MAX Plus Thermo Scientific™ Vanquish™ UHPLC system equipped with
a Thermo Scientific™ TriPlus™ RSH autosampler, following the method described by Kourtchev et al. (2022).
Only PFAS detected above the method's LOD (Kourtchev et al., 2022) or above blank levels were considered in the data
interpretation. All data were corrected for chamber dilution to account for the continuous inflow of clean air required to
maintain stable chamber pressure. No correction was applied for potential chamber wall losses. Because the chamber and
associated lines are made of stainless steel, and long-chain PFAS are known to interact strongly with metal surfaces, additional
losses to the walls are likely and may contribute to the overall uncertainty. However, as all experiments were conducted under
the same conditions, any wall-loss effects are expected to be systematic and should not affect the relative comparison of the
results.

## 2.5 Multiple-Path Particle Dosimetry (MPPD) modelling

For each region of the human respiratory tract (Head, Tracheobronchial region (TB) and Pulmonary region(P), the deposition
efficiency as a function of particle size, was estimated using the MPPD model (version 2.11, Applied Research Associates,
Inc.) (Anjilvel & Asgharian, 1995), using the default human breathing parameters (further details and results are provided in





the supplement file). The MPPD model is based on a framework of semiempirical equations and computational algorithms
that simulates particle deposition in the respiratory tract using anatomical and physiological data. It accounts for species-
specific airway geometry, breathing patterns, and particle characteristics to estimate regional and total deposition under various
exposure scenarios.
The deposition flux, $DF_{r,i}$ (pg h⁻¹) to each region of the human respiratory tract (Head, TB or Pulmonary), $r$, for each
nanoMOUDI size bin, $i$ (assuming constant exposure), was then calculated as

$$DF_{r,i} = DE_{r,i} \times C_i \times V$$

where $DE_{r,i}$ is the deposition efficiency, $C_i$ (pg m⁻³) is the measured concentration of the target PFAS and $V$ (m³ h⁻¹) is the
human breathing rate. The deposition fluxes for the coarse (> 2.5 µm), fine (0.1 – 2.5 µm), and ultrafine (< 0.1µm) size
fractions were then calculated by summing across the relevant MOUDI size bins.

**3 Results and Discussions**

**3.1 Total suspended particles (TSP)**

Figure 1 shows concentration of individual PFAS measured in the TSP fraction for the PFAS-only (Exp 1–3) and PFAS with
*Pseudomonas fluorescens* seeded (Exp 4–6) experiments.
Distinct trends were observed between the sulfonated and carboxylated PFAS. The sulfonated compounds, i.e. PFBS, PFHxS,
PFOS, and the fluorotelomer sulfonates (4:2, 6:2, and 8:2 FTS), exhibited relatively uniform aerosol-phase concentrations,
indicating that chain length had little influence on their aerosolisation efficiency. Their consistent behaviour likely reflects the
inherently high surface activity of sulfonates, which promotes their enrichment at the air water interface (Klevan et al., 2025;
Lyu et al., 2022).
In contrast, the PFCA showed a clear chain-length dependence, with aerosol-phase concentrations increasing from PFBA to
PFUdA. In a 40:60 methanol-water system, methanol decreases surface tension and solvent polarity, enhancing the solubility
and mobility of longer-chain PFCA relative to pure water (Kutsuna et al., 2012). This mixed-solvent environment therefore
favours the transfer of hydrophobic carboxylates into the aerosol phase. It must be noted that short-chain PFAS e.g. PFBA
(C₃) are less surface active and can remain in solution (Cai et al., 2022; Klevan et al., 2025).
Accordingly, sulfonated PFAS appear dominated by interfacial adsorption, whereas carboxylated PFAS are more strongly
affected by bulk-phase solvation governed by solvent composition. While direct comparison with earlier studies is limited by
methodological differences, similar behaviour has been observed during water aeration, where perfluoro sulfonated
compounds exhibited higher aerosolisation efficiencies than carboxylated analogues (Pandamkulangara Kizhakkethil et al.,
2024; Pandamkulangara Kizhakkethil and Kourtchev, 2025). A similar trend, involving an increase in aerosol-phase
perfluorinated alkyl acids, their salts and conjugate bases abundance with perfluoroalkyl chain length but not equivalent





enrichment magnitudes, was also observed under highly aqueous (tap water) conditions in bubble-bursting experiments using
a plunging jet, which is considered representative of nascent sea spray aerosol formation, as reported by Reth et al. (2011).
In the presence of *Pseudomonas fluorescens* (Fig. 1, Exp 4–6), the overall TSP concentrations of most PFAS were comparable
to those in the PFAS-only experiments, indicating that bacterial seeds did not substantially influence PFAS aerosolisation
under the tested conditions. The average and standard deviation of bacteria concentration across the 3 replicated experiments,
measured by WIBS, 5 minutes after the end of injection was $29 \pm 1$ # cm$^{-3}$. Slight reductions observed for some long-chain
PFCA (e.g. PFNA, PFDA, PFUdA) may reflect weak sorptive interactions with bacterial cell walls or their fragments, although
these effects appear minor relative to the dominant physicochemical controls. The overlap of standard deviations between the
two experimental conditions (with and without *Pseudomonas fluorescens*) suggests that PFAS concentrations in aerosol were
similar within the experimental uncertainty.
It must be noted that the major fraction of *Pseudomonas fluorescens* present in the chamber was observed around 0.6 μm
(Figure 2), smaller than the typical bacterial dimension (about 2-4 μm in length and 0.5-1.0 μm in diameter). It is worth noting
that the nebulisation processes exert stress on bacteria, producing fragmentation (Park et al., 2009). Particles in this size range
lie within the accumulation mode and therefore may act as efficient condensation sinks for condensable species (Engvall et
al., 2008). In the present experiments, PFAS were introduced in ionic form via nebulisation, yielding PFAS in both the gas
phase and the particle phase (the latter associated with nebulised droplets and their dried residues). Any enrichment of PFAS
on the pre-existing bacterial particles would therefore have required gas–particle partitioning to the bacterial surface or
particle–particle interactions such as coagulation. The absence of measurable enhancement in PFAS aerosol concentrations
therefore suggests that condensation or adsorption of PFAS onto bacterial surfaces was not thermodynamically favourable, or
that kinetic limitations prevented significant mass transfer during the experimental timescale. This suggests that, under the
applied conditions, PFAS aerosol formation and growth were predominantly governed by nebulisation and subsequent droplet
drying processes rather than by heterogeneous uptake onto biological particles.

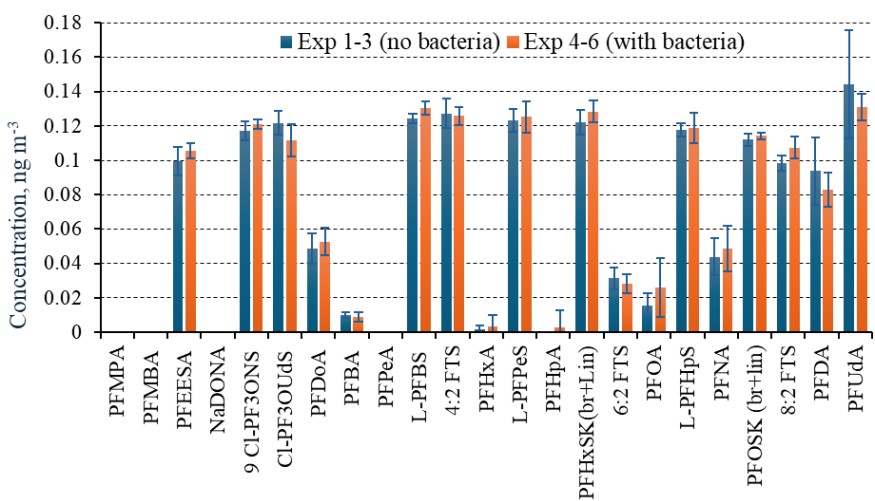






**Figure 1: Aerosol-phase concentrations of individual PFAS measured in total suspended particles (TSP) during PFAS-only**
**experiments (Exp 1–3, blue) and PFAS +** *Pseudomonas fluorescens*–**seeded experiments (Exp 4–6, orange). Error bars show the**
**standard deviation across three chamber replicates with duplicate LC–MS analyses per replicate (n = 6).**

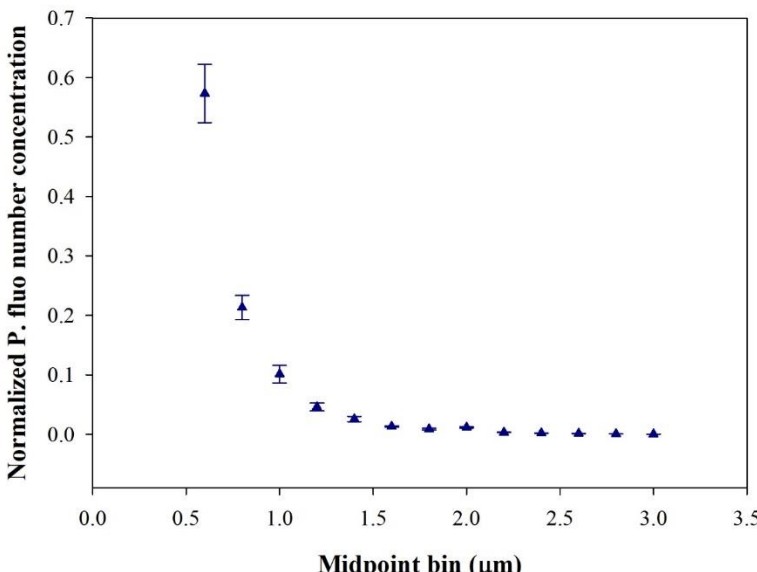


**Figure 2: Normalised particle size distribution of** *Pseudomonas fluorescens* **nebulised in ChAMBRe (5 minutes after the end of**
**bacteria nebulisation). Data represent the mean ± standard deviation of three replicate experiments.**
**3.2 Mass–Size Distributions of PFAS**
As mentioned in the Methodology section, not all PFAS detected in the TSP samples were also observed in the NanoMOUDI
samples, with carboxylated PFAS being mainly affected. This was likely due to their lower aerosol concentrations in the
chamber (as also observed in TSP) and their generally higher volatility (e.g. PFBA, 6.37 mmHg at 25 °C, Steele et al., 2002;
PFHxA, 13 mmHg at 25 °C, US EPA, 2012), which reduce particle-phase partitioning. Additionally, "dilution" across multiple
MOUDI stages may have further contributed to concentrations falling below the LC-MS detection limit in the NanoMOUDI
samples.
The majority of PFAS (excluding 6:2 FTS and PFOS) aerosols exhibited a consistent unimodal mass–size distribution, peaking
at 0.32 μm (Figure 3). Interestingly, variations in the molecular composition of the tested PFAS, including differences in
perfluorocarbon chain length and terminal functional group, did not affect the mode diameter of the aerosol mass–size
distribution. PFAS, representing a broad range of compounds, differ markedly in hydrophobicity and interfacial activity as a
function of both chain length and functional group (e.g. Lyu et al., 2022; Patel et al., 2024; Leung et al., 2023), which makes
the present observation somewhat unexpected. For instance, long-chain sulfonates such as PFOS (C8) exhibit considerably
stronger surface activity than short-chain carboxylates such as PFBA (C3), and their behaviour in bulk aqueous systems differs
accordingly (Guo et al., 2023).





This suggests that, under the applied experimental conditions, aerosol formation and size characteristics were largely governed
by physical processes. The most plausible explanation is that the aerosol-generation method imposed uniform physical
constraints during droplet formation and solvent evaporation, limiting the extent to which molecular properties influenced
particle characteristics. In addition, the use of an organic solvent likely enhanced the solubility of all PFAS, including long-
chain species with low water solubility, allowing them to remain in solution and aerosolise more uniformly during nebulisation.
Methanol substantially reduces surface tension (from 71.7 dyne cm$^{-1}$ for pure water to 38.7 dyne cm$^{-1}$ at 40 % v/v methanol
at 25 °C; Cheong and Carr, 1987), which promotes droplet formation and minimises differences in surface activity among
PFAS, thereby obscuring potential molecular-specific effects on aerosol behaviour.
The two analytes, 6:2 FTS and PFOS, did not follow the general mass-size distribution trend. Although PFOS exhibited a
dominant submicron mass mode at 0.32 µm, similar to other PFAS, its mass-size profile showed a relatively larger fraction of
mass in the smallest measured bins. In other words, PFOS retained the common 0.32-µm residual peak but also displayed
enrichment in the ultrafine fraction compared with other PFAS in the mixture. PFOS may produce a larger fraction of ultrafine
aerosol particles than other PFAS due to its greater surface activity (Klevan et al., 2025; Lyu et al., 2022). Although the
presence of methanol substantially reduces bulk surface tension, PFOS can still dominate dynamic interfacial processes during
rapid droplet formation. Its strong and persistent adsorption at the air–liquid interface, combined with enrichment as methanol
evaporates, likely lowers local surface tension further and inhibits coalescence, resulting in smaller and more stable droplets.
In addition, the anionic nature of PFOS may contribute to electrostatic stabilisation of charged droplets, further enhancing the
ultrafine fraction.
Other long-chain PFAS in the mixture did not exhibit similar enrichment, potentially due to competitive adsorption and
intermolecular interactions in mixtures that modulate their effective surface activity.
Only a limited number of studies have reported size-resolved mass distributions of PFAS associated with atmospheric particles
from the field observations. Comparison with previous work shows that PFAS size distributions vary considerably among
studies. Harada et al. (2006) and Dreyer et al. (2015) reported compound-dependent patterns, with PFOA and other PFCA
enriched in fine or ultrafine particles, whereas PFOS tended to occur in coarser fractions. In contrast, Guo et al. (2018) found
both PFOA and PFOS primarily associated with fine particles (< 1 µm), while Ge et al. (2017) observed PFCA in ultrafine
particles (< 0.1 µm) and PFOS and other sulfonates in coarse modes. Such variability likely reflects differences in sources,
atmospheric conditions, and sampling methodologies, as well as local physicochemical environments influencing PFAS
partitioning.
To the best of our knowledge, previous laboratory investigations of PFAS aerosol size behaviour have focused primarily on
sea-spray systems, which are not directly comparable to the organic solvent-rich aerosolisation process examined here.
Johansson et al. (2019) reported that the highest enrichment of perfluoroalkyl acids (PFAA) relative to seawater occurred in
aerosols with aerodynamic diameters below 1.6 µm. Sha et al. (2021) found that particle surface-area-to-volume ratio was a
strong predictor of PFAS enrichment in supermicron particles but not in submicron particles, indicating that different physical
controls operate across size ranges. In their subsequent work, Sha et al. (2024) observed that PFAS enrichment was particularly



pronounced in submicrometer sea-spray aerosol particles and varied with chain length and dissolved organic matter content.
These studies suggest that in marine systems, PFAS enrichment and size association are sensitive to experimental conditions
and molecular structure, with both particle-scale physics and surfactant properties influencing partitioning. In contrast, the
consistent submicron unimodal distribution observed in the present work likely reflects aerosol formation under organic-
solvent conditions, where rapid solvent evaporation and solute concentration effects impose dominant physical constraints that
reduce the influence of PFAS molecular features on particle size.
The introduction of *Pseudomonas fluorescens* into the chamber as a potential seed or carrier of PFAS, in most cases, had either
no effect (within experimental error) or resulted in a uniform decrease in PFAS aerosol mass size concentration across nearly
the entire size range for two analytes (i.e. PFEESA and GenX), while the distribution profile and modal diameter remained
unchanged. As mentioned in the section 3.1, the major fraction of *Pseudomonas fluorescens* in the chamber was observed
around 0.6 µm (Fig 2). The absence of PFAS enrichment at the bacterial modal size (~0.6 µm) and the unchanged PFAS modal
diameter (~0.3 µm) indicate that *Pseudomonas fluorescens* did not noticeably influence PFAS size distribution under the tested
conditions. This was somewhat unexpected, as PFAS have been shown to associate with bacterial surfaces in aqueous systems
(Dai et al., 2023). The physicochemical environment in aerosols likely may differ from that in bulk water. The outer membrane
of *Pseudomonas fluorescens* carries a net negative charge, arising from acidic functional groups on lipopolysaccharides and
phospholipids (Boyd Chelsea et al., 2014; Charlton et al., 2024), while the tested PFAS are also anionic. Such electrostatic
repulsion could therefore further inhibit PFAS attachment, potentially explaining the absence of observable enrichment at ~0.6
µm. In addition, although PFAS are amphiphilic, their molecular structure makes the air–water interface far more favourable
for stabilisation than the hydrated, negatively charged bacterial surface, and PFAS therefore likely preferentially stabilise at
droplet interfaces rather than adsorb onto bacterial cells. Another aspect worth considering is whether *Pseudomonas*
*fluorescens* could have influenced PFAS aerosol concentrations through biochemical transformation rather than solely through
physical carrier processes.  In bulk aqueous systems, several Pseudomonas species have been reported to partially degrade
sulfonated PFAS, particularly precursors such as H-PFOS, under nutrient-enriched or co-metabolic conditions (e.g., Key et
al., 1998). The latter work involved liquid culture media with high bacterial densities, organic carbon co-substrates, and
prolonged incubation times, which facilitate enzymatic activity and redox transformations. However, even in that study,
evidence for complete degradation of PFOS is lacking; rather, transformation is slow, partial, and often requires co-metabolic
drivers. By contrast, the conditions in our aerosol chamber differ fundamentally. The bacteria were suspended in air with
transient water content (RH~40%), rather than immersed in nutrient-rich aqueous media. Under such conditions, the metabolic
activity of *Pseudomonas fluorescens* is expected to be extremely limited. The observed uniform decrease in PFAS aerosol
mass concentration cannot therefore be straightforwardly attributed to microbial degradation, as the air–water interface-
dominated microenvironment is unlikely to sustain enzymatic pathways known to act on PFAS in bulk liquid cultures.
Furthermore, the residence time of particles in the chamber is orders of magnitude shorter than the timescales over which
reported PFAS transformations by Pseudomonas occur (days to weeks). It is therefore more plausible that the apparent decrease
in GenX and PFEESA aerosol mass concentrations reflect non-biological processes such as redistribution of material to



chamber or sampler walls, or surface-competition dynamics during condensation, rather than direct microbial influence. If any
biochemical contribution occurred, it would likely be negligible compared with these physicochemical pathways. The
difference in TSP concentrations between the *Pseudomonas fluorescens*-seeded (average 0.105±0.0043 ng m-3, n=3) and
unseeded (0.099±0.008 ng m-3, n=3) experiments for PFEESA was minimal, suggesting that the concentration drop observed
in the NanoMOUDI size-resolved data likely resulted from sampling artefacts or volatility-driven size redistribution rather
than bacterial activity. In this respect, it has been shown that compounds with higher volatility tend to exhibit greater mass
losses through evaporation particularly in impactor-based sampling systems like NanoMOUDI (e.g. Ungeheuer et al., 2022).
Although the same inferences could not be made for GenX from the TSP data, due to high background levels in the TSP
blanks, evaporation from the collection substrate in the NanoMOUDI was likely the dominant loss process, as also suggested
for PFEESA. The relatively high vapour pressure of GenX (2.7 mm Hg at 20 °C, US EPA, 2022) supports this interpretation.
In previous work, it was found that the culturable lifetime of P. fluorescens in ChAMBRe in dark condition was about 20
minutes (Gatta et al. 2025). Furthermore, the survival of bacteria in air is known to be sensitive to aerosolisation and sampling
conditions rather than simply liquid-phase growth (Després et al., 2012; Hong et al., 2021). It must be noted that in our study
the bacteria were seeded in sterilised MQ, whereas PFAS were introduced separately via a 40:60 (v/v) methanol-water
nebulisation. Under this experimental setup, bacteria were not subjected to high alcohol strength in the droplet phase; instead,
their methanol exposure was dominated by chamber-average vapour and sporadic interactions near the spray plume. These
conditions are unlikely to produce strong biocidal effects at the population level, so the absence of PFAS enrichment at the
bacterial size mode is better explained by interfacial/partitioning constraints than by methanol-induced loss of viability.
Moreover, even if a fraction of the bacterial population experienced viability loss, this would not preclude potential PFAS
interactions with biological surfaces. Non-viable bacterial cells, cell-wall fragments, and microbial biomass retain abundant
functional groups (carboxyl, phosphate, and amine moieties) known to sorb organic and inorganic species (Fathollahi & Coupe,
2021; Torres, 2020; Wang & Chen, 2009). Inactivated bacterial biomass is widely used as a biosorbent due to preserved surface
chemistry and polymeric matrices (Torres 2020). In the atmosphere, biological particles occur not only as intact cells but also
as cell fragments and exudates (Després et al., 2012; Fröhlich-Nowoisky et al., 2016), meaning that surface area and chemical
functionality persist even when viability is compromised. Thus, even partial loss of viability would still permit association of
PFAS with microbial surfaces if interfacial partitioning were favourable. The lack of detectable PFAS at the bacterial size
mode therefore reinforces that limited affinity/partitioning, rather than loss of cellular integrity, governed PFAS–bioaerosol
interactions under our experimental conditions.









**Figure 3: Mass size distribution of aerosol from PFAS-only (blue symbols) and PFAS with *Pseudomonas fluorescens* seed (red symbols) experiments.**

**3.3 Multiple-Path Particle Dosimetry (MPPD) modelling**

Figure 4 (with data given in the supplement) presents the deposition fluxes across respiratory regions (head, TB, pulmonary) and the relative contributions of inhaled PFAS associated with coarse, fine, and ultrafine particles. The majority of PFAS measured exhibited similar size distributions, with most particles falling within the fine fraction (0.1 – 2.5 µm), resulting in broadly consistent deposition patterns. The average total deposition flux across these PFAS was $5.3 \pm 1.0$ pg h$^{-1}$, with deposition distributed relatively evenly between the head ($23\% \pm 2\%$, 1.22 pg h$^{-1}$), TB ($26\% \pm 1\%$, 1.34 pg h$^{-1}$), and pulmonary ($40\% \pm 1\%$, 2.08 pg h$^{-1}$) regions.

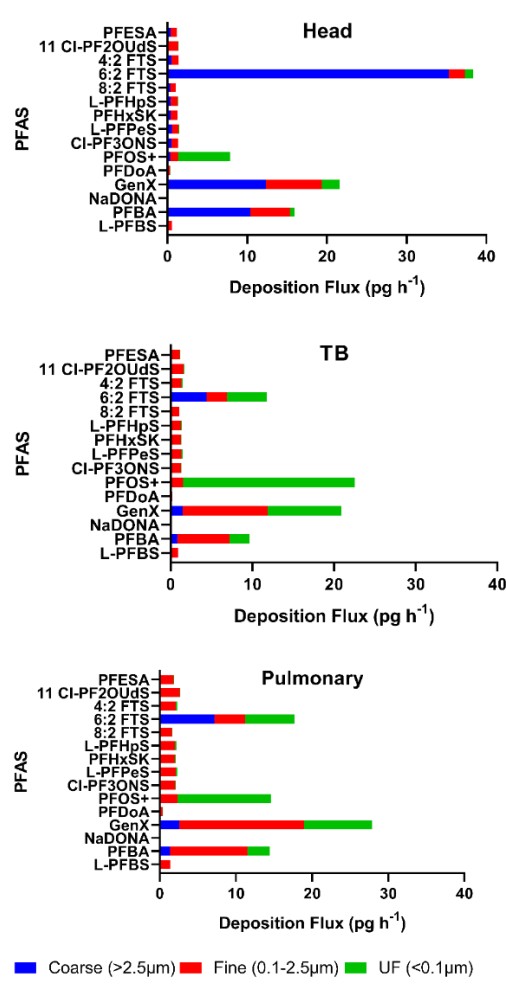

**Figure 4: Modelled fractional deposition flux (pg h$^{-1}$) in the Pulmonary, Tracheobronchial (TB) and Head regions of the human respiratory tract.**





PFDoA and NaDONA were notable exceptions due to their significantly lower aerosol concentrations, resulting in total deposition fluxes of 1.18 pg h$^{-1}$ and 0.138 pg h$^{-1}$, respectively. PFDoA's size distribution was similar to the majority, yielding comparable regional deposition: 29% to the head (0.34 pg h$^{-1}$), 22% to the TB region (0.26 pg h$^{-1}$), and 36% to the pulmonary region (0.43 pg h$^{-1}$). NaDONA however exhibited an additional peak in the coarse fraction, shifting deposition toward the head region, which received 57% of the total flux (0.08 pg h$^{-1}$), while the TB and pulmonary regions received 13% (0.02 pg h$^{-1}$) and 21% (0.03 pg h$^{-1}$), respectively. PFOS and PFBA showed significantly higher aerosol concentrations, with total deposition fluxes of 48.1 pg h$^{-1}$ and 44.3 pg h$^{-1}$, respectively. PFBA had a similar size distribution to the majority but at higher concentration, resulting in deposition fractions of 36% to the head (15.9 pg h$^{-1}$), 22% to the TB region (9.7 pg h$^{-1}$), and 33% to the pulmonary region (14.5 pg h$^{-1}$). PFOS, however, had an additional peak in the ultrafine range, shifting deposition toward the TB region, which received 47% of the total flux (22.5 pg h$^{-1}$), while the head region received 16% (7.9 pg h$^{-1}$). GenX and 6:2 FTS exhibited the highest aerosol concentrations and deposition fluxes, at 77.5 pg/h and 74.4 pg h$^{-1}$, respectively. GenX followed the typical deposition pattern, with 28% to the head (21.6 pg h$^{-1}$), 27% to the TB region (20.9 pg h$^{-1}$), and 36% to the pulmonary region (27.9 pg h$^{-1}$). In contrast, 6:2 FTS had additional modes in both the ultrafine and coarse fractions, leading to a deposition profile skewed toward the head region, which received 52% of the total flux (38.4 pg/h), while the TB and pulmonary regions received 16% (11.8 pg h$^{-1}$) and 24% (17.7 pg h$^{-1}$), respectively. This pattern was similar to that observed for NaDONA.

Particle deposition in the respiratory tract is primarily governed by inertial impaction, gravitational sedimentation, and Brownian diffusion, each dominating in different regions depending on particle size and airflow. Coarse particles deposit mainly in the upper airways via impaction and sedimentation, while ultrafine particles reach the distal pulmonary region through diffusion. Modelled deposition efficiencies by size and region are provided in the SI. Comparing the relative contribution to the deposition flux of the inhaled PFAS associated with the different size fractions, coarse particles showed the highest deposition flux in the head region (60% ± 3%), with lower contributions to the pulmonary (15% ± 2%) and tracheobronchial (3% ± 8%) regions. Fine particles (0.1–2.5 µm), which dominate the PFAS size distribution, had the lowest overall deposition efficiency but were more evenly distributed: 39% ± 10% to the pulmonary region, 25% ± 7% to TB, and 18% ± 5% to the head. Ultrafine particles shifted deposition toward the pulmonary region (45% ± 14%), with 35% ± 11% to TB and only 8% ± 3% to the head.

The modelled deposition behaviour of PFAS compounds investigated is closely linked to their particle size distribution. PFOS, GenX, 6:2 FTS, and PFBA exhibited significant ultrafine fractions, suggesting a higher likelihood of deep lung penetration. This is particularly concerning given that clearance mechanisms in the pulmonary region are slower compared to upper airways, and pulmonary deposition increases the potential for translocation into the bloodstream. These findings highlight the importance of considering both particle size and regional deposition when assessing inhalation exposure risks, especially for compounds with known toxicological profiles and environmental persistence.




## 4 Conclusions

This study examined the aerosol formation and size-resolved distribution of a range of PFAS under controlled chamber conditions, using mixed water–organic systems with and without the model bacterium *Pseudomonas fluorescens* to assess the influence of molecular structure, interfacial behaviour and biological material on aerosol properties.

In terms of aerosolisation efficiency, sulfonated PFAS exhibited broadly similar aerosol-phase concentrations across chain lengths, whereas perfluoroalkyl carboxylic acids showed increasing aerosolisation with increasing chain length, highlighting the influence of functional group and hydrophobicity on the overall transfer of PFAS into the particle phase.

Most PFAS were associated with the fine aerosol mode, displaying unimodal mass–size distributions centred near 0.3 µm. This consistent fine-mode behaviour across PFAS of differing chain lengths and functional groups indicates that aerosol formation was governed primarily by physical processes of droplet generation and evaporation. PFOS showed enhanced ultrafine enrichment, while 6:2 FTS and NaDONA displayed broader profiles, suggesting that differences in volatility and interfacial behaviour introduce secondary compound-specific effects.

The presence of *Pseudomonas fluorescens* as an aerosol seed did not enhance PFAS aerosolisation or alter modal diameters, but resulted in small, compound-specific reductions, particularly for PFEESA and GenX. The absence of PFAS enrichment at the bacterial modal diameter (~0.6 µm) indicates limited association of PFAS with bacterial surfaces under the tested conditions, likely reflecting electrostatic repulsion and preferential stabilisation of PFAS at air–liquid interfaces. These results suggest that biological material exerts only a minor influence on PFAS partitioning through the airborne pathway examined here; however, aqueous-phase sorption or complexation before aerosolisation may still contribute to water-to-air transfer and warrants further investigation. Moreover, if similar behaviour holds in the atmosphere, surface-active PFAS may avoid shifting into bioaerosol particle sizes with higher deposition velocities and therefore remain in the fine aerosol range with longer atmospheric lifetimes and transport potential.

MPPD simulations using the experimental size distributions indicated that most aerosol-bound PFAS would deposit in the pulmonary region. Compounds with stronger ultrafine enrichment, including PFOS, 6:2 FTS and GenX, showed higher predicted deposition in distal lung regions where clearance is slow and transfer into epithelial lining fluids is more likely. Under the studied conditions, PFAS-containing aerosols therefore fall largely within respirable size ranges relevant to inhalation exposure.

Overall, PFAS aerosolisation and particle-phase behaviour in the mixed solvent system were dominated by the physical processes of droplet formation and evaporation, suggesting that engineering and operational measures that suppress fine droplet production could reduce airborne PFAS emissions. The observed fine-mode distributions also imply that aerosolised PFAS may be efficiently transported in the atmosphere and contribute to inhalation exposure beyond immediate emission sources. Although the water–methanol system does not fully reflect environmental conditions, it provides a controlled basis for identifying the fundamental processes governing PFAS transfer from contaminated aqueous systems to air.



Future work should apply this framework to more environmentally representative matrices, including natural organic matter, and diverse microbial assemblages, to better capture real emission complexity. Combining controlled chamber experiments with field measurements of size-resolved PFAS and bioaerosol emissions will be essential for improving predictions of PFAS atmospheric transport, deposition and human exposure.

**5 Financial Support**

This work was supported by the European Commission, Horizon 2020 Research Infrastructures (EUROCHAMP-2020) Trans National Access grant ATMO-TNA-6M-0000000034, "Investigation of the aerosolisation dynamics and biological interactions of GenX "forever chemicals" – AEROBIOGEN". The authors gratefully acknowledge the Coventry University QR funding for supporting the Trailblazers PhD studentship awarded to JPK, secured by IK, and also thank CAWR at Coventry University for its financial assistance.

**6 Competing Interests**

Some authors are members of the editorial board of journal ACP. The authors have no other competing interests to declare.

**7 Author Contributions**

IK, SC, FM, EG, DM, PP conceived the study. IK, JPK, SC, FM, DM, EG, VV performed lab measurements, sampling, and sample analysis. IK, JPK, FM, EG performed data processing and interpretation. AB performed modelling. IK, JPK, FM, AB prepared the original draft of the paper. All authors contributed to reviewing and editing the manuscript.

**8 Data Availability**

The dataset for this work can be accessed at DOI 10.5281/zenodo.17756209

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
