# Peer review of "(PFAS) aerosols with Pseudomonas fluorescens: size distributions,"

_EGUsphere, 2025_

## Referee Comment (RC1)

**Major comments**

**1. External vs. Internal Mixing and Environmental Relevance**

A key concern is that the mixing state between bacteria and PFAS in the chamber experiments differs fundamentally from the environmental scenario described in the Introduction.

Bacteria and PFAS were aerosolized separately using different generators, resulting in external mixing in the chamber air. In contrast, wastewater treatment plant aeration produces internally mixed aerosols because microorganisms and PFAS are already mixed in the aqueous phase prior to bubble bursting.

Given the low particle concentrations and short residence times in chamber studies, the probability of meaningful interaction between separately generated bacterial aerosols and PFAS droplets in air is very low. Therefore, the reported lack of bacterial influence on PFAS aerosol formation may be a consequence of the experimental design rather than a true absence of interaction.

The authors should more clearly acknowledge this limitation and restrict the scope of their conclusions, or modify the experimental setup to better represent internal mixing conditions relevant to wastewater environments.

**2. Wall loss**

Wall loss is likely to vary as a function of PFAS chain length, yet the manuscript appears to assume that relative comparisons remain valid because all compounds experience wall loss. It is not clear that this assumption is justified. If wall-loss efficiencies differ systematically with chain length, then the observed trends may be influenced, at least in part, by differential losses rather than intrinsic behavior.

Therefore, in sections where chain-length-dependent trends are discussed, this limitation should be explicitly acknowledged and its potential impact on the interpretation of the results should be clearly addressed.

**3. MPPD modeling**

The MPPD modeling results are inherently driven by particle size. According to the manuscript, particle size in this study was strongly influenced by experimental conditions

such as solvent composition and the use of a nebulizer. Given this, it is unclear whether interpreting the modeled deposition results on a compound-specific basis is appropriate.

Moreover, while the authors acknowledge that the experimental setup does not fully represent real environmental conditions, it remains questionable whether predicting the magnitude and regional distribution of respiratory deposition is truly meaningful or environmentally representative. The limitations associated with applying the MPPD model to aerosol size distributions generated under these experimental constraints should therefore be explicitly discussed, and the interpretation of the modeling results should be framed accordingly.

**Minor comments**

1. Line 85-99

The current discussion appears to rely heavily on listing information rather than synthesizing it. Reorganizing and more clearly integrating the key points would help condense the content and improve overall readability.

2. Line 127

Please check the superscripts.

3. Line 170

Please check whether 109 cells/mL is correct.

4. Line 156 and 174

Both L/min and lpm are used in the manuscript. It would be preferable to use a single, consistent unit throughout.

5. Figure 1

For readers who are not familiar with PFAS abbreviations, it may be difficult to infer chain length from compound names alone. You may want to consider reorganizing the

compounds by grouping them into the same classes (e.g., PFCAs, PFSAs) and arranging them in order of chain length within each group to improve clarity and readability.

6. Figures

Please consider expanding the figure caption slightly. At present, the caption mainly describes what is shown in the figure. Adding one sentence that highlights the key takeaway or main message of the figure would help readers better understand its significance.

---

## Author Comment (AC6)

We thank Reviewer 3 for their comments and suggestions. We have carefully considered these and provide point-by-point responses below. Responses are given in blue, with cited or newly added text indicated in *italic*.

**Reviewer 3**

General comments

PFAS are present in atmospheric aerosols; however, their interfacial chemistry at aerosol surfaces remains poorly understood. This study investigated the behaviour of PFAS in aerosols in both the absence and presence of Pseudomonas fluorescens as a biological seed. By illustrating a unimodal aerosol PFAS mass distribution, the study showed association of PFAS with the fine particle mode. No effect of Pseudomonas fluorescens addition on aerosol PFAS concentration was observed. The study applied Multiple-Path Particle Dosimetry (MPPD) modeling, based on the measured particle size distributions, to evaluate particle deposition in the respiratory tract.

 **Response**: We thank the reviewer for this summary of the study.

**Research comments (Reviewer 3):** The PFAS-containing mixture introduced into the chamber would be expected to exhibit compound-specific wall deposition, which should be monitored prior to the start of the experiment. Were wall loss deposition rates measured? Stainless steel chambers are known to exhibit significant wall deposition for organic acids and peroxides; could this have affected the experiments? Please comment in detail on the possible contribution of wall losses in relation to compound vapor pressure.

**Response:** We would like to thank the reviewer for this comment. Wall losses were not measured and as stated in the original submission not applied for data correction (lines 214-215). Direct quantification of PFAS aerosol wall losses would require time-resolved measurements of aerosol-phase PFAS concentrations throughout the chamber residence time. This is not technically feasible for PFAS at slightly higher than environmentally relevant concentrations used in our experiments, as no online measurement techniques are available to quantify ionic PFAS in the aerosol phase in the lab. Offline determination of wall losses would therefore necessitate repeated sampling at multiple time points to resolve concentration gradients within the chamber. Considering the low aerosol-phase PFAS concentrations obtained from short, sequential filter sampling, such an approach would introduce substantial analytical uncertainty.

Using higher PFAS concentrations to quantify wall losses is also problematic, as wall-loss processes are known to be concentration- and size dependant (Wang et al., 2018), such that results obtained under elevated concentrations are not necessarily representative of losses potentially occurring in our experiments.

To reflect on this point, we have added a statement to the manuscript acknowledging the inability to directly quantify wall losses and clarifying that, while wall losses may influence absolute recoveries, the reported trends represent conservative estimates of aerosol-phase PFAS behaviour under the conditions studied:

> "*Direct quantification of aerosol-phase PFAS wall losses was not performed, as it would require time-resolved aerosol-phase PFAS measurements that are not technically feasible at the concentrations applied, given the absence of online*

*measurement techniques for ionic PFAS and the need for extended offline sampling to achieve sufficient analytical sensitivity, especially in size-resolved samples.*" Lines 228-231, revised manuscript.

We also acknowledged limitations and support of the approach resulting from this phenomenon:

"*Moreover, wall-loss efficiencies may vary with PFAS chain length and could influence absolute recoveries. All experiments were therefore conducted under consistent chamber configuration and operating conditions to minimise variability in wall-loss behaviour and to allow relative comparisons within a common experimental framework.*" Lines 233-236, revised document.

"*While a clear chain-length-dependent increase in aerosol-phase PFCA concentrations is observed, any preferential wall losses of longer-chain compounds would act to reduce their measured recovery and therefore bias the magnitude of this increase towards lower values*." Lines 264-266, revised document.

**Reviewer 3 comment:** How was dilution due to aerosol sampling corrected? What was the dilution rate during the experiments?

**Response:** Dilution due to aerosol sampling was accounted for by correcting the measured concentrations using the total sampling and dilution flow rates applied during the experiment. The corresponding clarification has now been added to the SI section:

"*Dilution due to aerosol sampling was corrected by accounting for the total sampling flow rate relative to the chamber volume, treating sampling as a first-order loss process. The total volume of ChAMBRe is 2,200 L. The instruments flow was:*

- *NanoMOUDI 10 L min$^{-1}$;*

- *TSP: 10 L min$^{-1}$;*

- *OPS: 1 L min$^{-1}$;*

- *SMPS: 1 L min$^{-1}$;*

- *WIBS: 0.3 L min$^{-1}$;*

*In the experiments with only PFAS, the instruments used were MOUDI, TSP, OPS and SMPS. The dilution factor was 12/2200 min$^{-1}$ = 0.01 min$^{-1}$.*

*In the experiments with PFAS + bacteria, the instruments used were MOUDI, TSP, OPS, SMPS and WIBS. The dilution factor was 12.3/2200 min$^{-1}$ = 0.0101 min$^{-1}$.*"

**Reviewer 3 comment:** The nebulization process may exert mechanical stress and cause fragmentation of bacteria; however, is this the only explanation for the observed fragmentation of Pseudomonas fluorescens? Could the processes differ under higher relative humidity (RH) conditions?

**Response:** The observed fragmentation of *Pseudomonas fluorescens* is consistent with mechanical stress induced during nebulisation; however, this is unlikely to be the only contributing mechanism. Rapid droplet evaporation at the moderate RH (~40%) used in this study likely imposed additional desiccation and osmotic stress, promoting further

fragmentation of already weakened cells. Under higher RH conditions, slower evaporation would be expected to reduce desiccation stress and preserve a larger fraction of intact cells or aggregates, leading to a shift toward larger particle sizes. Thus, bacterial fragmentation reflects a combination of mechanical and humidity-dependent physicochemical processes rather than nebulisation alone.

**Reviewer 3 comment:** Smaller aerosol particles experience higher internal pressure due to curvature effects enhanced when PFAS are present. How might this influence the mass accommodation of Pseudomonas fluorescens onto or into PFAS-containing aerosols?

**Response:** Curvature-related (Kelvin) effects are primarily relevant for very small (nm) droplets, diminishing rapidly as particle size increases beyond the tens of nanometres range (e.g. Tröstl et al., 2016). In contrast, *Pseudomonas fluorescens* cells are µm sized (reported cell length 1–2 µm; width 0.3–0.6 µm), placing them several orders of magnitude larger than the droplet sizes for which Kelvin effects are significant and at least an order of magnitude larger than the fine aerosol particles dominating the PFAS mass distribution. Furthermore, mass accommodation coefficients are defined for molecular uptake and describe the probability of gas-phase molecules entering the condensed phase, rather than interactions involving aerosolised PFAS and intact biological cells. Accordingly, curvature-induced internal pressure effects are not expected to control interactions between *Pseudomonas fluorescens* and PFAS-containing aerosols; any influence of PFAS is more plausibly related to changes in surface properties rather than Kelvin-driven accommodation.

**Reviewer 3 comment:** PFAS-containing aerosols may be stable at smaller diameters depending on RH. How would RH affect the reported results? Did the authors perform experiments at different RH levels, and were similar conclusions obtained? Higher RH is expected to promote stronger adhesion, whereas lower RH may result in more reversible interactions.

**Response:** Although relative humidity can influence aerosol water content and phase state, the present results remain important because they demonstrate that biological aerosol seeding does not measurably enhance PFAS aerosolisation or alter size-resolved PFAS mass distributions under moderate relative humidity conditions (∼40%), which are representative of many atmospheric environments. The absence of a detectable biological effect at this RH places a meaningful constraint on the role of bioaerosols in PFAS atmospheric partitioning, indicating that any potential RH-dependent enhancement would need to be substantial to alter the conclusions reported here.

A systematic assessment of relative humidity and temperature effects is indeed of interest and represents a valuable avenue for further investigation; however, such an approach would require analysis of a very large number of samples due to the 13-stage size-resolved MOUDI sampling, the need for experimental replication, and the inclusion of additional test conditions, with each condition generating at least 39 samples. At an analytical cost of approximately £180 per sample, this was not feasible within the scope of the present study, but remains an important direction for future research.

To acknowledge this point, the following statement has been added to the Conclusions:

> "*Variability in temperature and relative humidity may influence PFAS aerosol behaviour through effects on aerosol water content and evaporation dynamics and should be*

*considered when extrapolating these findings to broader atmospheric conditions.*"
Lines 502-504, revised document.

**Minor comments**

**Reviewer 3 comment:** Line 51: Please check whether "and" should be used instead of "&" for ACP.

Response: Corrected to "*and*". Thank you.

**Reviewer 3 comment:** Line 138: Consider moving the PFAS mixture description to the Supplementary Information.

Response: We agree with the reviewer that detailed compound lists should be kept concise where possible. For this reason, only the PFAS details required to interpret the Results and Discussion are listed in the main text, while the names of isotopically labelled compounds that are not discussed later were already placed in the Supplementary Information. The remaining mixture description is retained in the main text to allow readers to follow compound-specific results without repeated cross-referencing and interrupting the narrative flow, particularly given the number of analytes and the use of abbreviations throughout.

**Reviewer 3 comment:** Line 174: The unit for liter is given as "L" in the text; please check the use of "lpm."

Response: Changed to L min$^{-1}$

**Reviewer 3 comment:** Line 197: Please provide information regarding the extraction time and storage period.

Response: Although this information is not directly relevant to the long-term stability of PFAS as "forever chemicals", details on storage duration and extraction timing have now been added to the text, indicating storage for ~ 60 days prior to analysis and extraction and analysis within 24 hours per batch:

> "*All samples were analysed within 24 hours of extraction for each analytical batch.*" Lines 224-225, revised manuscript.

> "*..approximately for 60 days at −20 °C until extraction.*" Line 210, revised document.

**References:**

Tröstl, J., Chuang, W. K., Gordon, H., Heinritzi, M., Yan, C., Molteni, U., Ahlm, L., Frege, C., Bianchi, F., Wagner, R., Simon, M., Lehtipalo, K., Williamson, C., Craven, J. S., Duplissy, J., Adamov, A., Almeida, J., Bernhammer, A.-K., Breitenlechner, M., Brilke, S., Dias, A., Ehrhart, S., Flagan, R. C., Franchin, A., Fuchs, C., Guida, R., Gysel, M., Hansel, A., Hoyle, C. R., Jokinen, T., Junninen, H., Kangasluoma, J., Keskinen, H., Kim, J., Krapf, M., Kürten, A., Laaksonen, A., Lawler, M., Leiminger, M., Mathot, S., Möhler, O., Nieminen, T., Onnela, A., Petäjä, T., Piel, F. M., Miettinen, P., Rissanen, M. P., Rondo, L., Sarnela, N., Schobesberger, S., Sengupta, K., Sipilä, M., Smith, J. N., Steiner, G., Tomè, A., Virtanen, A., Wagner, A. C., Weingartner, E., Wimmer, D., Winkler, P. M., Ye, P., Carslaw, K. S., Curtius, J., Dommen, J.,

Kirkby, J., Kulmala, M., Riipinen, I., Worsnop, D. R., Donahue, N. M., and Baltensperger, U.: The role of low-volatility organic compounds in initial particle growth in the atmosphere, Nature, 533, 527–531, https://doi.org/10.1038/nature18271, 2016.

Wang, N., Jorga, S. D., Pierce, J. R., Donahue, N. M., and Pandis, S. N.: Particle wall-loss correction methods in smog chamber experiments, Atmos. Meas. Tech., 11, 6577–6588, https://doi.org/10.5194/amt-11-6577-2018, 2018.